# Robust Pseudo-label Learning with Neighbor Relation for Unsupervised Visible-Infrared Person Re-Identification

Xiangbo Yin*
School of Informatics,
Xiamen University
Xiamen, China
xiangboyin@stu.xmu.edu.cn

Jiangming Shi*
Institute of Artificial Intelligence,
Xiamen University
Xiamen, China

Yachao Zhang
Tsinghua Shenzhen International
Graduate School, Tsinghua University
Shenzhen, China

Yang Lu
Key Laboratory of Multimedia
Trusted Perception and Efficient
Computing, Ministry of Education of
China, Xiamen University
Xiamen, China

Zhizhong Zhang
School of Computer Science
and Technology,
East China Normal University
Shanghai, China

Yuan Xie†
East China Normal University
Shanghai, China
Chongqing Institute of East China
Normal University,
Chongqing, China

Yanyun Qu†
Key Laboratory of Multimedia
Trusted Perception and Efficient
Computing, Ministry of Education of
China, Xiamen University
Xiamen, China
yyqu@xmu.edu.cn

## Abstract

Unsupervised Visible-Infrared Person Re-identification (USVI-ReID) presents a formidable challenge, which aims to match pedestrian images across visible and infrared modalities without any annotations. Recently, clustered pseudo-label methods have become predominant in USVI-ReID, although the inherent noise in pseudo-labels presents a significant obstacle. Most existing works primarily focus on shielding the model from the harmful effects of noise, neglecting to calibrate noisy pseudo-labels usually associated with hard samples, which will compromise the robustness of the model. To address this issue, we design a Robust Pseudo-label Learning with Neighbor Relation (RPNR) framework for USVI-ReID. To be specific, we first introduce a straightforward yet potent Noisy Pseudo-label Calibration module to correct noisy pseudo-labels. Due to the high intra-class variations, noisy pseudo-labels are difficult to calibrate completely. Therefore, we introduce a Neighbor Relation Learning module to reduce high intra-class variations by modeling potential interactions between all samples. Subsequently, we devise an Optimal Transport Prototype Matching module to establish reliable cross-modality correspondences. On that basis, we design a Memory Hybrid Learning module to jointly learn modality-specific and modality-invariant information. Comprehensive experiments conducted on two widely recognized benchmarks, SYSU-MM01 and RegDB, demonstrate that RPNR outperforms the current state-of-the-art GUR with an average Rank-1 improvement of 10.3%. The code is available at https://github.com/XiangboYin/RPNR.

## CCS Concepts

• **Computing methodologies → Object identification**.

## Keywords

USVI-ReID, Noisy Labels, Neighbor Relation Learning, Optimal Transport

**ACM Reference Format:**

Xiangbo Yin, Jiangming Shi, Yachao Zhang, Yang Lu, Zhizhong Zhang, Yuan Xie, and Yanyun Qu. 2024. Robust Pseudo-label Learning with Neighbor Relation for Unsupervised Visible-Infrared Person Re-Identification. In *Proceedings of the 32nd ACM International Conference on Multimedia (MM'24)*. October 28-November 1, 2024, Melbourne, VIC, Australia, 10 pages. https://doi.org/10.1145/3664647.3680951

*Equal Contribution.
†Corresponding author.

## 1 Introduction

With the increasing demand for intelligent security, smart monitoring sensor devices for 24-hour surveillance are becoming more prevalent [21, 31, 34, 52, 58]. Due to the different imaging principles of sensor devices during the daytime and nighttime, the data exhibits multi-modality characteristics, sparking interest in

research on visible-infrared person re-identification (VI-ReID). VI-ReID aims to accurately search the special visible/infrared pedestrian images when given a query pedestrian image from another modality, however, the significant gap between the two modalities presents a considerable challenge for this task. Recently, a series of VI-ReID methods [12, 13, 18, 48, 53, 60, 61] have aimed to reduce cross-modality discrepancies by aligning visible and infrared images at both the image and feature levels, resulting in significant performance improvements. However, these methods rely on well-annotated cross-modality data, which is time-consuming and labor-intensive in practical scenarios. Therefore, increasing attention is being drawn to unsupervised visible-infrared person re-identification (USVI-ReID).

The key challenges of the USVI-ReID are obtaining robust pseudo-labels and establishing reliable cross-modality correspondences. Existing USVI-ReID methods [4, 29, 45, 49] mostly follow the DCL [51] framework, which generates pseudo-labels using DBSCAN and establishes cross-modality correspondences based on pseudo-labels. Since pseudo-labels are generated by clustering, they inevitably contain noise. The noisy pseudo-labels may cause the model to incorrectly learn the data distributions and feature representations. To mitigate the effects of the noisy pseudo-labels, DPIS [32] computes the confidence scores of pseudo-labels by analyzing their classifier loss, then uses confidence scores to mitigate the impact of noisy pseudo-labels. PGM [45] reduces the impact of noisy labels by alternately using two unidirectional metric losses, preventing the rapid formation of noisy pseudo-labels. However, these methods don't calibrate noisy pseudo-labels to clear ones, which makes it difficult for the model to exploit hard-to-discriminate features. In order to establish cross-modality correspondences, OTLA [39] utilizes unsupervised domain adaptation to generate pseudo-labels for the infrared images. With the aid of richly annotated visible images, it proposes an optimal transport strategy to allocate pseudo-labels from the visible modality to the infrared modality. However, OTLA adopts the strategy of independently assigning pseudo-labels to each infrared image, which is a massive task with many distractors, leading to unreliable cross-modality correspondences.

In this paper, we propose the Robust Pseudo-Label Learning with Neighbor Relation (RPNR) framework, a unified approach aimed at addressing noisy pseudo-labels and cross-modality correspondences for USVI-ReID. To be specific, to calibrate noisy pseudo-labels, we design two critical modules: Noisy Pseudo-label Calibration (NPC) and Neighbor Relation Learning (NRL). Unlike previous methods that only reduce the effect of noisy pseudo-labels, NPC directly calibrates them. NPC obtains robust prototypes through reliable neighbor samples and calibrates pseudo-labels based on similarity to these prototypes. The significant intra-class variations will hinder the noisy pseudo-label calibration. NRL is proposed to reduce intra-class variations by interacting across all images. NRL promotes the model to cluster closely with neighbor samples, as neighbor samples are often related. In order to establish reliable cross-modality correspondences, we also design two critical modules: Optimal Transport Prototype Matching (OTPM) and Memory Hybrid Learning (MHL). Unlike OTLA, which overlooks the intra-class information and treats all images as separate instances for establishing cross-modality correspondences, OTPM employs intra-class information to build cross-modality correspondences. In brief,

OTPM obtains the prototype by clustering and establishes cross-modality correspondences based on these prototypes instead of all instances. Furthermore, the significant cross-modality gaps will hinder the establishment of cross-modality correspondences. MHL is designed to learn both modality-specific and modality-invariant information by blending two modality-specific memories, effectively bridging the substantial gaps between different modalities.

In conclusion, the main contributions of our method can be summarized as follows:

- We propose the Robust Pseudo-Label Learning with Neighbor Relation (RPNR) framework to address both noisy pseudo-labels and noisy cross-modality correspondence problems in USVI-ReID.
- Two critical modules: Noisy Pseudo-label Calibration (NPC) and Neighbor Relation Learning (NRL) are introduced to obtain robust pseudo-labels.
- Two critical modules: Optimal Transport Prototype Matching (OTPM) and Memory Hybrid Learning (MHL) are introduced to establish reliable cross-modality correspondences.
- Experiments on SYSU-MM01 and RegDB datasets demonstrate the superiority of our method compared with existing USVI-ReID methods, and RPNR generates higher-quality pseudo-labels than other methods.

## 2 Related Work

### 2.1 Unsupervised Single-Modality Person ReID

Unsupervised single-modality person ReID aims to learn discriminative identity features from unlabeled person ReID datasets. Existing mainstream purely unsupervised methods primarily rely on pseudo-labels, which involve an iterative process alternating between pseudo-label generation and representation learning [6, 9, 10, 14, 24, 36, 37, 40, 41, 63, 67]. Cluster-Contrst [8, 20] presents a cluster contrast framework, which stores unique centroid representations and performs contrastive learning at the cluster level. Additionally, the momentum update strategy is introduced to reinforce the cluster-level feature consistency in the embedding space. However, a uni-proxy for a cluster may introduce bias. To address this issue, multi-proxies methods [44, 66] have been proposed to compensate for the shortcomings of uni-proxy approaches. Pseudo-labels inherently contain a portion of noise. To address this issue, label refinement methods [5, 6, 62] are proposed to collect more reliable pseudo-labels. While the aforementioned methods have shown promising results in unsupervised ReID tasks, applying them directly to unsupervised VI-ReID scenarios poses a significant challenge due to the substantial cross-modality gap.

### 2.2 Unsupervised Visible-Infrared Person ReID

There has been an escalating interest in unsupervised visible-infrared person re-identification (USVI-ReID) owing to its potential to learn modality-specific and modality-invariant information without necessitating cross-modality annotations. Existing mainstream USVI-ReID methods [4, 17, 28, 30, 45, 46, 49, 50] predominantly adhere to the DCL [51] learning framework, which involves two key steps: (1) generating pseudo-labels using a clustering algorithm, and (2) establishing cross-modality correspondences based on these pseudo-labels. PGM [45] and MBCCM [17] perform multi-stage graph

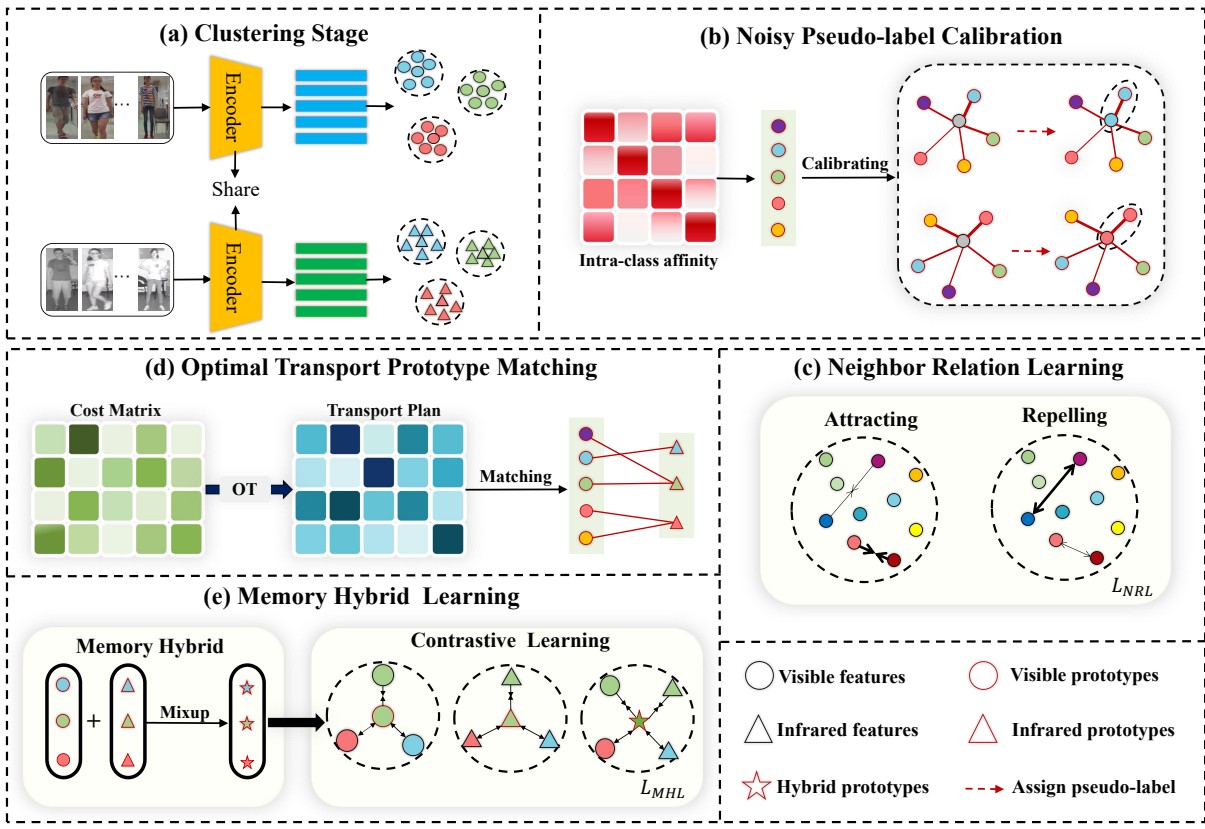

**Figure 1: Overall of the proposed RPNR. RPNR first generates modality-specific pseudo-labels by DBSCAN at stage (a). After that, RPNR calibrates noisy pseudo-labels (grey dots) to obtain robust pseudo-labels (color dots) at stage (b), while modeling potential interactions between all samples (the strength is indicated by thickness) to reduce intra-class variation at stage (c). Additionally, based on the robust pseudo-labels, RPNR employs the optimal transport to establish cross-modality correspondences at stage (d). Finally, with the recast-aligned pseudo-labels, RPNR mixes two modality-specific memories as a new hybrid memory to learn both modality-specific and modality-invariant information through contrastive loss at stage (e).**

matching via building bipartite graphs. OTLA [39] and DOTLA [4] employ the Optimal Transport strategy to assign pseudo-labels from one modality to another modality at the instance level. However, pseudo-labels inevitably contain noise, which may lead to unreliable cross-modality correspondences under the supervision of noisy pseudo-labels. Therefore, there is a need to seek more reliable pseudo-labels for the USVI-ReID task.

### 2.3 Learning with Noisy Labels

The presence of label noise has been demonstrated to adversely affect the training of deep neural networks [35, 54, 55]. Existing methods devised for handling noisy labels can be primarily categorized into the following two classes: label correction and sample selection. Label correction methods [2, 33, 38, 65] endeavor to utilize the model predictions to rectify the noisy labels. [16] proposes an iterative learning framework SMP to relabel the noisy samples and train the network on the real noisy dataset without using extra clean supervision. [59] utilizes back-propagation to probabilistically update and correct image labels beyond updating the

network parameters. Different from label correction methods, sample selection methods [15, 23, 42] aim to select clean samples while discarding noisy samples during the training stage. NCE [22] filters the clean samples according to the neighbor information. CBS [25] proposes to employ confidence-based sample augmentation to enhance the reliability of selected clean samples. For the USVI-ReID task, pseudo-labels generated by the clustering algorithm inevitably contain noise. Therefore, calibrating these noisy pseudo-labels is crucial for improving the performance of USVI-ReID.

## 3 Methodology

### 3.1 Notation Definition

Given an unlabeled visible-infrared person re-identification dataset $\mathcal{D} = \{\mathcal{D}^{\mathcal{V}}, \mathcal{D}^{\mathcal{R}}\}$, where $\mathcal{D}^{\mathcal{V}} = \{x_i^v \mid i = 1, 2, \ldots, N^v\}$ represents the unlabeled visible dataset with $N^v$ samples and $\mathcal{D}^{\mathcal{R}} = \{x_i^r \mid i = 1, 2, \ldots, N^r\}$ represents the unlabeled infrared dataset with $N^r$ samples. For the USVI-ReID task, the objective is to train a robust model $f_\theta$ to project a sample $x_i^t$ from $\mathcal{D}$ into an embedding space $\mathcal{F}$, where $t = \{v, r\}$. Thus, we can employ the encoder $f_\theta$ to extract

visible features $F^v = \{f_i^v \mid i = 1, 2, \ldots, N^v\}$ and infrared features $F^r = \{f_i^r \mid i = 1, 2, \ldots, N^r\}$, where $f_i^t \in \mathbb{R}^d$.

## 3.2 Overview

The overall framework of the proposed method is shown in Fig. 1. We first employ the DBSCAN [11] algorithm to cluster visible and infrared features. After clustering, we can obtain pseudo-labels $y_i^t \in \{1, 2, \ldots, Y^t\}$ of $i$-th images from modality $t$, where $Y^t$ is the number of clusters and $t = \{v, r\}$. Since pseudo-labels inevitably contain noise, we first propose a Noisy Pseudo-label Calibration (NPC) module to calibrate noisy labels to obtain more robust pseudo-labels. Afterward, we assign these calibrated pseudo-labels $\hat{y}_i^t$ for each sample to obtain the "labeled" dataset $\tilde{\mathcal{D}}^V = \{(x_i^v, \hat{y}_i^v)\}_{i=1}^{N^v}$ and $\tilde{\mathcal{D}}^R = \{(x_i^r, \hat{y}_i^r)\}_{i=1}^{N^r}$. However, NPC overlooks the possible interactions between all samples. To make up for this deficiency, we propose a Neighbor Relation Learning (NRL) module, which is designed to model the intricate interactions spanning across all samples. Furthermore, the pseudo-labels generated by two separate clustering for visible and infrared samples reveal a misalignment. To align correspondences between visible and infrared samples, we design an Optimal Transport Prototype Matching (OTPM) module, which considers cross-modality correspondences as alignment between visible and infrared prototypes by optimal transport. Learning modality-invariant features is crucial in cross-modality correspondences. To better mine the modality-invariant information and alleviate significant cross-modality gaps, we propose a Memory Hybrid Learning (MHL) module, which mixes aligned visible and infrared prototypes as new modality-hybrid prototypes for contrastive learning.

## 3.3 Noisy Pseudo-label Calibration

Since pseudo-labels are generated by clustering, they inevitably contain noise. We introduce the Noisy Pseudo-label Calibration (NPC) module to correct noisy pseudo-labels. Specifically, given the $c$-th cluster from modality $t$, corresponding to a set of $d$-dimensional features $\{f_{c,i}^t\}_{i=1}^{n_c}$, where $n_c$ denotes the number of features belonging to $c$-th cluster and $t \in \{v, r\}$. We employ the *Jaccard Similarity* to model the affinity matrix $\mathcal{S}$ of intra-class samples as follows:

$$S_{ij}^t = \frac{\left| \mathcal{R}\left(f_{c,i}^t, \kappa\right) \cap \mathcal{R}\left(f_{c,j}^t, \kappa\right) \right|}{\left| \mathcal{R}\left(f_{c,i}^t, \kappa\right) \cup \mathcal{R}\left(f_{c,j}^t, \kappa\right) \right|}, \tag{1}$$

where $S_{ij}^t$ is the affinity between $f_{c,i}^t$ and $f_{c,j}^t$, and $\mathcal{R}\left(f_{c,i}^t, \kappa\right)$ is the $\kappa$-reciprocal nearest neighbors of $f_{c,i}^t$. The larger $S_{ij}^t$, the higher similarity between $f_{c,i}^t$ and $f_{c,j}^t$. For a specific $f_{c,i}^t$, if it is surrounded by more similar samples, the sample is more likely to be reliable. To select reliable samples for a cluster, we design a Similarity Counter $G_c^t$ for each sample:

$$G_{c,i}^t = \sum_{j=1}^{n_c} sign(S_{ij}^t - \rho), i \in \{1, 2, \ldots, n_c\}, \tag{2}$$

where $sign(\cdot)$ is a *sign* function and $\rho$ is a threshold fixed to 0.5. We can find that correctly categorized samples should have higher similarity counts, so we regard the samples with the top-$K$ similarity

counts as reliable samples:

$$id = \arg\max_K G_c^t, \tag{3}$$

where $id$ denotes the indexes of top-$K$ similarity counts.

Then we can obtain a robust prototype with these reliable samples for the $c$-th cluster:

$$p_c^t = \frac{1}{K} \sum_{i \in id} f_{c,i}^t. \tag{4}$$

After that, we can own a prototype set $p^t = \{p_1^t, p_2^t, \ldots, p_{Y^t}^t\}$. For a given sample $x_i^t$ from $\mathcal{D}^t$, the similarity score $\delta_{c,i}^t$ between the extracted feature $f_i^t$ and the $c$-th cluster is calculated as follows:

$$\delta_{c,i}^t = \frac{(f_i^t) \cdot (p_c^t)^T}{\|f_i^t\|_2 \cdot \|p_c^t\|_2}, \tag{5}$$

where $\delta_{c,i}^t$ denotes the cosine similarity between the extracted feature $f_i^t$ and the $c$-th cluster. Larger $\delta_{c,i}^t$ indicates the sample $x_i^t$ is more likely to belong to the $c$-th cluster. Then, we can obtain the corrected pseudo-label by:

$$\hat{y}_i^t = \arg\max_c \delta_{c,i}^t, c \in \{1, 2, \ldots, Y^t\}. \tag{6}$$

Afterward, we assign these corrected labels for each sample for network training.

## 3.4 Neighbor Relation Learning

Considering high intra-class variability profoundly hampers the NPC module, we propose the Neighbor Relation Learning (NRL) module, which is designed to reduce intra-class variability through the intricate interactions spanning across all pair-wise samples. Following [19], we employ Relaxed Contrastive Loss for learning semantic embedding of pair-wise samples. For convenience, we only explain the process of visible samples. Given a pair of samples $(f_i^v, f_j^v)$, we compute the Euclidean distance between them by:

$$d_{ij}^v = \left\| f_i^v - f_j^v \right\|_2. \tag{7}$$

Then, the visible loss of the NRL module can be formulated:

$$L_{NRL}^v = \underbrace{\frac{1}{N_B} \sum_{i=1}^{N_B} \sum_{j=1}^{N_B} \omega_{ij}^v d_{ij}^{v\,2}}_{\text{attracting}} + \underbrace{\frac{1}{N_B} \sum_{i=1}^{N_B} \sum_{j=1}^{N_B} (1 - \omega_{ij}^v)[\gamma - d_{ij}^v]_+^2}_{\text{repelling}}, \tag{8}$$

where $N_B$ denotes the number of samples in each iteration and $\gamma$ is a margin hyper-parameter. $[x]_+$ denotes $\max(0, x)$, which is a hinge function. Moreover, $\omega_{ij}^v$ is the weight term, formulated by a Gaussian kernel function based on the Euclidean distance:

$$\omega_{ij}^v = \exp\left(-\frac{\|f_i^v - f_j^v\|_2^2}{\sigma}\right), \tag{9}$$

where $\sigma$ represents the kernel bandwidth and $\omega_{ij}^v \in (0, 1]$. Obviously, it can be used to measure the similarity relation of paired samples in the embedding space.

As shown in Eq. (8), the NRL loss contains an attracting term and a repelling term. The positive pairs will approach each other with

the help of the attracting term and the repelling term encourages the negative pairs to push away beyond the margin $\gamma$.

Similarly, the NRL loss of the infrared modality is defined as:

$$L_{NRL}^r = \underbrace{\frac{1}{N_B} \sum_{i=1}^{N_B} \sum_{j=1}^{N_B} \omega_{ij}^r {d_{ij}^r}^2}_{\text{attracting}} + \underbrace{\frac{1}{N_B} \sum_{i=1}^{N_B} \sum_{j=1}^{N_B} (1 - \omega_{ij}^r)[\gamma - d_{ij}^r]_+^2}_{\text{repelling}}.$$

(10)

Therefore, the total loss of the NRL module is:

$$L_{NRL} = L_{NRL}^v + L_{NRL}^r. \tag{11}$$

## 3.5 Optimal Transport Prototype Matching

The two modules mentioned above primarily concentrate on intra-modality information while overlooking inter-modality connections, which are pivotal in the USVI-ReID task. To this end, following PGM [45] and OTLA [39], we present the Optimal Transport Prototype Matching (OTPM) module to establish reliable cross-modality correspondences at the cluster level. Given the visible prototype set $p^v = \{p_1^v, p_2^v, \ldots, p_{Y^v}^v\}$ and the infrared prototype set $p^r = \{p_1^r, p_2^r, \ldots, p_{Y^r}^r\}$, where $Y^v$ and $Y^r$ represent the number of visible clusters and infrared clusters, respectively. PGM revealed $Y^v > Y^r$, that is, the number of clusters is inconsistent. In that case, the essence of cross-modality correspondences is the many-to-many matching of inter-modality prototypes, which can be solved by *Optimal Transport*:

$$\min_Q \langle Q, C \rangle + \frac{1}{\lambda}\mathcal{H}(Q),$$
$$\text{s.t.} \begin{cases} Q\mathbb{1} = \mathbb{1} \cdot \frac{1}{Y^v}, \\ Q^T\mathbb{1} = \mathbb{1} \cdot \frac{1}{Y^r}, \end{cases}$$

(12)

where $Q \in \mathbb{R}^{Y^v \times Y^r}$ represents the transport plan for cross-modality matching. $C \in \mathbb{R}^{Y^v \times Y^r}$ is the cost matrix of inter-modality prototypes, i.e., $C_{ij} = 1/\exp\left(\cos\left(p_i^v, p_j^r\right)\right)$, where $\cos(\cdot)$ indicates the cosine similarity. $\langle \cdot \rangle$ denotes the Frobenius dot-product, and $\mathbb{1}$ is an all-one vector. $\mathcal{H}(Q)$ denotes the Entropic Regularization and $\lambda$ is a regularization parameter. The objective function can be solved by the Sinkhorn-Knopp algorithm [7] and derive the optimal transport plan $Q^* \in \mathbb{R}^{Y^v \times Y^r}$. Then we can obtain two matched pseudo-label sets $Y^{v \to r}$ and $Y^{r \to v}$ for network training according to $Q^*$:

$$Y_i^{v \to r} = \arg\max_j Q_{ij}^*, j \in \{1, 2, \ldots, Y^r\},$$
$$Y_j^{r \to v} = \arg\max_i Q_{ji}^*, i \in \{1, 2, \ldots, Y^v\}.$$

(13)

## 3.6 Memory Hybrid Learning

We initialize two modality-specific memory banks $\mathcal{M}^v \in \mathbb{R}^{Y^v \times d}$ and $\mathcal{M}^r \in \mathbb{R}^{Y^r \times d}$ by clustering centroids. However, the two modality-specific memories only store the modality-specific information, which can't mine modality-invariant information and reduce the cross-modality discrepancy. To this end, with the cross-modality correspondences derived from the OTPM, we propose a Memory Hybrid Learning (MHL) module to jointly learn modality-specific information and modality-invariant information.

Firstly, we create a modality-hybrid memory $\mathcal{M}^h \in \mathbb{R}^{Y^r \times d}$ to store modality-shared information via mixing matched visible and

infrared prototypes by:

$$p_i^h = \alpha \times p_i^r + (1 - \alpha) \times p_i^{r \to v},$$
$$\mathcal{M}_i^h \leftarrow p_i^h,$$

(14)

where $i \in \{1, 2, \ldots, Y^r\}$ and $p_i^{r \to v}$ denotes the visible prototype which matches with the infrared prototype $p_i^r$. $\alpha$ is a balancing hyper-parameter that balances the fusion information of the visible and infrared prototypes.

Afterward, during the representation learning stage, we follow the popular memory-based methods [4, 17, 45, 49, 51], which mainly alternate two stages: (1) performing contrastive learning during forward-propagation (FP) and (2) updating the memory during backward-propagation (BP). To better learn the representations, we perform multi-memory joint contrastive learning, which consists of modality-specific contrastive learning and modality-invariant contrastive learning.

**Modality-Specific Contrastive Learning.** Based on the modality-specific memory $\mathcal{M}^v$ and $\mathcal{M}^r$, the ClusterNCE [8] loss is applied to learn modality-specific information for network optimization by:

$$L_{MS}^v = -\frac{1}{N_B} \sum_{i=1}^{N_B} \log \frac{\exp\left(f_i^v \cdot \mathcal{M}^v[\hat{y}_i^v]/\tau\right)}{\sum_{k=1}^{Y^v} \exp\left(f_i^v \cdot \mathcal{M}^v[\hat{y}_k^v]/\tau\right)}, \tag{15}$$

$$L_{MS}^r = -\frac{1}{N_B} \sum_{j=1}^{N_B} \log \frac{\exp\left(f_j^r \cdot \mathcal{M}^r[\hat{y}_j^r]/\tau\right)}{\sum_{k=1}^{Y^r} \exp\left(f_j^r \cdot \mathcal{M}^r[\hat{y}_k^r]/\tau\right)}, \tag{16}$$

where $N_B$ denotes the number of samples in each iteration. $\hat{y}_i^v$ and $\hat{y}_j^r$ are the pseudo-labels of query features $f_i^v$ and $f_j^r$. $\mathcal{M}^v[\hat{y}_i^v]$ and $\mathcal{M}^r[\hat{y}_j^r]$ denote the positive representations of query features $f_i^v$ and $f_j^r$, respectively. Besides, $\tau$ is a temperature hyper-parameter. The total loss of modality-specific contrastive learning is:

$$L_{MS} = L_{MS}^v + L_{MS}^r. \tag{17}$$

During the backward-propagation stage, the two modality-specific memories are updated by a momentum update strategy:

$$\mathcal{M}^v[\hat{y}_i^v] \leftarrow \mu\mathcal{M}^v[\hat{y}_i^v] + (1 - \mu)f_i^v, \tag{18}$$

$$\mathcal{M}^r[\hat{y}_j^r] \leftarrow \mu\mathcal{M}^r[\hat{y}_j^r] + (1 - \mu)f_j^r, \tag{19}$$

where $\mu$ is the momentum updating factor.

**Modality-Invariant Contrastive Learning.** Different from two modality-specific memories, we perform modality-invariant contrastive learning on modality-shared memory $\mathcal{M}^h$ to learn modality-invariant information while reducing the cross-modality discrepancy. Following PGM [45], we employ the alternate contrastive learning scheme on $\mathcal{M}^h$:

$$L_{MI} = \begin{cases} -\frac{1}{N_B} \sum_{i=1}^{N_B} \log \frac{\exp\left(f_i^v \cdot \mathcal{M}^h[\hat{y}_i^{v \to r}]/\tau\right)}{\sum_{k=1}^{Y^r} \exp\left(f_i^v \cdot \mathcal{M}^h[\hat{y}_k^{v \to r}]/\tau\right)}, & \text{if } Epoch\%2 = 0, \\ -\frac{1}{N_B} \sum_{i=1}^{N_B} \log \frac{\exp\left(f_i^r \cdot \mathcal{M}^h[\hat{y}_i^r]/\tau\right)}{\sum_{k=1}^{Y^r} \exp\left(f_i^r \cdot \mathcal{M}^h[\hat{y}_k^r]/\tau\right)}, & \text{if } Epoch\%2 = 1, \end{cases}$$

(20)

where $\hat{y}_i^{v \to r}$ denotes the visible pseudo-label $\hat{y}_i^v$ matched with the infrared pseudo-label $\hat{y}_i^r$. Then, the modality-shared memory is updated jointly by query features $f_i^v$ and $f_i^r$:

$$\mathcal{M}^h[\hat{y}_i^{v \to r}] \leftarrow \mu\mathcal{M}^v[\hat{y}_i^{v \to r}] + (1 - \mu)f_i^v, \quad \text{if } Epoch\%2 = 0,$$
$$\mathcal{M}^h[\hat{y}_i^r] \leftarrow \mu\mathcal{M}^r[\hat{y}_i^r] + (1 - \mu)f_i^r, \quad \text{if } Epoch\%2 = 1.$$

(21)

Table 1: Comparisons with state-of-the-art methods on RegDB and SYSU-MM01, including SVI-ReID, SSVI-ReID, and USVI-ReID methods. All methods are measured by Rank-1 (%) and mAP (%). GUR* denotes the results without camera information.

| Settings | | | RegDB | | | | SYSU-MM01 | | | |
| --- | --- | --- | --- | --- | --- | --- | --- | --- | --- | --- |
| | | | Visible2Thermal | | Thermal2Visible | | All Search | | Indoor Search | |
| Type | Method | Venue | Rank-1 | mAP | Rank-1 | mAP | Rank-1 | mAP | Rank-1 | mAP |
| SVI-ReID | DDAG [57] | ECCV'20 | 69.4 | 63.5 | 68.1 | 61.8 | 54.8 | 53.0 | 61.0 | 68.0 |
| | AGW [58] | TPAMI'21 | 70.1 | 66.4 | 70.5 | 65.9 | 47.5 | 47.7 | 54.2 | 63.0 |
| | CAJ [56] | ICCV'21 | 85.0 | 79.1 | 84.8 | 77.8 | 69.9 | 66.9 | 76.3 | 80.4 |
| | DART [52] | CVPR'22 | 83.6 | 75.7 | 82.0 | 73.8 | 68.7 | 66.3 | 72.5 | 78.2 |
| | LUPI [1] | ECCV'22 | 88.0 | 82.7 | 86.8 | 81.3 | 71.1 | 67.6 | 82.4 | 82.7 |
| | DEEN [64] | CVPR'23 | 91.1 | 85.1 | 89.5 | 83.4 | 74.7 | 71.8 | 80.3 | 83.3 |
| | PartMix [18] | CVPR'23 | 85.7 | 82.3 | 84.9 | 82.5 | 77.8 | 74.6 | 81.5 | 84.4 |
| | LCNL [53] | IJCV'24 | 85.6 | 78.7 | 84.0 | 76.9 | 70.2 | 68.0 | 76.2 | 80.3 |
| SSVI-ReID | OTLA [39] | ECCV'22 | 48.2 | 43.9 | 47.4 | 56.8 | 49.9 | 41.8 | 49.6 | 42.8 |
| | TAA [47] | TIP'23 | 62.2 | 56.0 | 63.8 | 56.5 | 48.8 | 42.3 | 50.1 | 56.0 |
| | DPIS [32] | ICCV'23 | 62.3 | 53.2 | 61.5 | 52.7 | 58.4 | 55.6 | 63.0 | 70.0 |
| USVI-ReID | OTLA [39] | ECCV'22 | 32.9 | 29.7 | 32.1 | 28.6 | 29.9 | 27.1 | 29.8 | 38.8 |
| | ADCA [51] | MM'22 | 67.2 | 64.1 | 68.5 | 63.8 | 45.5 | 42.7 | 50.6 | 59.1 |
| | CHCR [27] | TCSVT'23 | 68.2 | 63.8 | 70.0 | 65.9 | 47.7 | 45.3 | - | - |
| | DOTLA [4] | MM'23 | 85.6 | 76.7 | 82.9 | 75.0 | 50.4 | 47.4 | 53.5 | 61.7 |
| | MBCCM [17] | MM'23 | 83.8 | 77.9 | 82.8 | 76.7 | 53.1 | 48.2 | 55.2 | 62.0 |
| | CCLNet [3] | MM'23 | 69.9 | 65.5 | 70.2 | 66.7 | 54.0 | 50.2 | 56.7 | 65.1 |
| | PGM [45] | CVPR'23 | 69.5 | 65.4 | 69.9 | 65.2 | 57.3 | 51.8 | 56.2 | 62.7 |
| | GUR* [49] | ICCV'23 | 73.9 | 70.2 | 75.0 | 69.9 | 61.0 | 57.0 | 64.2 | 69.5 |
| | **RPNR** | - | **90.9** | **84.7** | **90.1** | **83.2** | **65.2** | **60.0** | **68.9** | **74.4** |

The total loss of the MHL module is:

$$L_{MHL} = L_{MS} + \beta_1 L_{MI}. \tag{22}$$

## 3.7 Optimization

The total training loss of the network can be formulated as follows:

$$L = L_{MS} + \beta_1 L_{MI} + \beta_2 L_{NRL}, \tag{23}$$

where $\beta_1$, $\beta_2$ are balancing coefficients, which are set to 0.5 and 10.0, respectively.

## 4 Experiment

## 4.1 Experiment Setting

**Datasets.** The proposed method is evaluated on two popular visible-infrared person re-identification datasets: **SYSU-MM01** [43] and **RegDB** [26]. More detailed explanations are presented in **Supplementary Materials**.

**Evaluation Metrics.** The experiment of our method was carried out following the evaluation metrics in DDAG [57], i.e., Cumulative Matching Characteristic (**CMC**) and Mean Average Precision (**mAP**). In the evaluation of our proposed method on the SYSU-MM01 dataset, we consider two distinct search modes: the All Search mode and the Indoor Search mode. Similarly, for the RegDB dataset, our method is evaluated across two testing modes: Visible2Thermal and Thermal2Visible.

**Implementation Details.** The proposed method is implemented on two TITAN RTX GPUs with PyTorch. During the training stage, all the input images are resized to 288×144, and data augmentations described in [56] are adopted for image augmentation. Following [58], we employ a two-stream feature extractor pre-trained on ImageNet to extract 2048-dimensional features. The number of training epochs is set to 100. The detailed settings are presented in **Supplementary Materials**.

## 4.2 Comparision with State-of-the-art Methods

**Comparison with SVI-ReID Methods.** Compared to SVI-ReID methods that rely on high-quality cross-modality annotations, the results of our RPNR are promising. As we can see, our method achieves comparable performance to some supervised methods (e.g., DDAG [57], AGW [58], and CAJ [56]), which is attributed to the fact that our proposed method can provide reliable pseudo-labels for unsupervised tasks.

**Comparison with SSVI-ReID Methods.** Several SSVI-ReID methods have been proposed to mitigate the issue of the high cost of cross-modality annotations. These methods utilize partial annotations to accomplish the VI-ReID task. It is noteworthy that our approach, without any cross-modality annotations, achieves a 6.8% improvement in Rank-1 and a 4.4% improvement in mAP on the SYSU-MM01 dataset compared to the SOTA DPIS method.

**Table 2: Ablation studies on the SYSU-MM01 dataset. Rank-R accuracy(%) and mAP(%) are reported.**

| | Module | | | | | All Search | | Indoor Search | |
|---|---|---|---|---|---|---|---|---|---|
| Order | Baseline | NPC | NRL | OTPM | MHL | Rank-1 | mAP | Rank-1 | mAP |
| 1 | ✓ | | | | | 40.4 | 39.0 | 42.3 | 51.2 |
| 2 | ✓ | ✓ | | | | 41.2 | 39.5 | 43.6 | 52.0 |
| 3 | ✓ | | ✓ | | | 42.5 | 41.4 | 45.5 | 53.9 |
| 4 | ✓ | ✓ | ✓ | | | 44.6 | 42.2 | 46.7 | 54.5 |
| 5 | ✓ | | | ✓ | ✓ | 60.2 | 55.8 | 62.9 | 69.3 |
| 6 | ✓ | ✓ | | ✓ | ✓ | 62.5 | 57.1 | 64.2 | 70.3 |
| 7 | ✓ | | ✓ | ✓ | ✓ | 63.7 | 57.4 | 64.7 | 70.5 |
| 8 | ✓ | ✓ | ✓ | ✓ | ✓ | 65.2 | 60.0 | 68.9 | 74.4 |

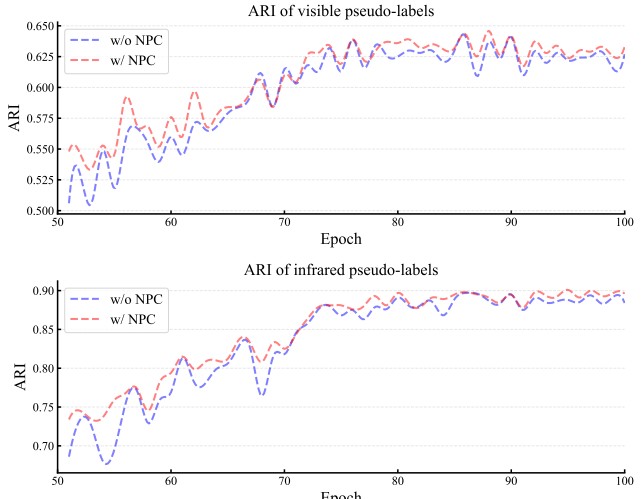

**Figure 2: The ARI metric of visible and infrared pseudo-labels on SYSU-MM01 at each epoch.**

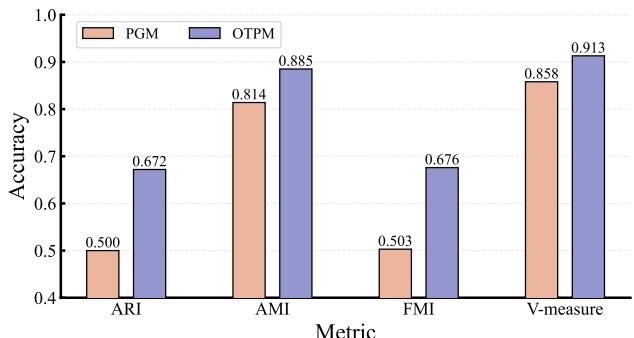

**Figure 3: The accuracy of cross-modality correspondences compared with PGM [45] on SYSU-MM01.**

**Comparison with USVI-ReID Methods.** As shown in Tab. 1, our method significantly outperforms existing SOTA USVI-ReID methods. To be specific, our RPNR achieves 65.2% in Rank-1 and 60.0% in mAP on SYSU-MM01, which surpasses GUR by 4.2% in Rank-1 and 3.0% in mAP. Surprisingly, the performance on RegDB achieves 90.9% in Rank-1 and 84.7% in mAP under the Visible2Thermal mode, which outperforms SOTA GUR by a large margin of 17.0% in Rank-1 and 14.5% in mAP. The results powerfully demonstrate the effectiveness of our approach, highlighting that our RPNR provides more reliable pseudo-labels and establishes more dependable cross-modality correspondences for USVI-ReID.

### 4.3 Ablation Study

To validate the effectiveness of each module in RPNR, we conduct ablation experiments on SYSU-MM01, as shown in Tab. 2. We employ the DCL framework with multiple proxies as the baseline.
**Effectiveness of the NPC Module.** The NPC module is proposed to explicitly rectify noisy pseudo-labels to obtain more reliable pseudo-labels. As shown in Order 5 and Order 6 in Tab. 2, the performance of Order 6 with NPC improves by about 2% compared

to Order 5. To more clearly demonstrate the effectiveness of the NPC module, we utilize the Adjusted Rand Index (ARI) metric to evaluate the accuracy of visible and infrared pseudo-labels on SYSU-MM01 at each epoch. A higher ARI value indicates more accurate pseudo-labels. As depicted in Fig. 2, the introduction of the NPC module results in improved accuracies for both visible and infrared pseudo-labels, thereby providing more reliable pseudo-labels for network training.
**Effectiveness of the NRL Module.** The NRL module is introduced as complementary information to make up for the shortcomings of rigid pseudo-labels. After adding the NRL module, the performance can gain improvement by 2%-4% in Rank-1 on SYSU-MM01. It shows that the NRL module can explore meaningful intricate interactions spanning across all pair-wise samples to provide complementary supervision information for the network.
**Effectiveness of the OTPM Module.** As shown in Fig. 3, we compared the cross-modality matching accuracy of OTPM with that of PGM on four clustering evaluation metrics to show the effectiveness of OTPM. As we can see, OTPM significantly outperforms the PGM on all four metrics, indicating its superior ability to establish reliable cross-modality correspondences at the cluster level.
**Effectiveness of the MHL Module.** We present the MHL module to jointly learn modality-specific and modality-invariant information while reducing cross-modality discrepancies. Note that the MHL module cannot be executed on its own, as it is built on top of the OTPM module. Compared to the Baseline, the combination of

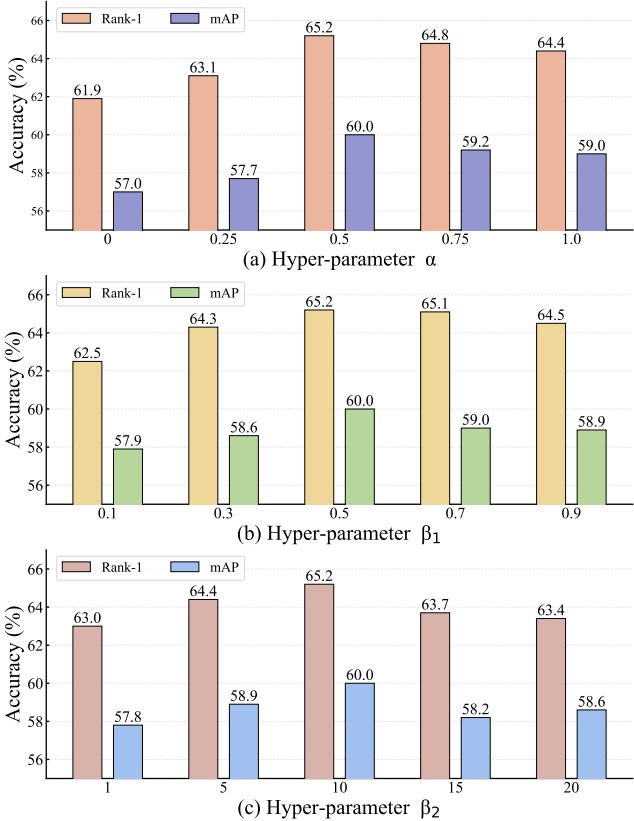

**Figure 4: The influence of three import hyper-parameters with different values on SYSU-MM01.**

MHL with OTPM leads to a significant performance improvement, with a large margin of 19.8% in Rank-1 accuracy and 16.8% in mAP (See Order 1 & Order 5). This highlights the efficiency of MHL in leveraging modality-specific and modality-invariant information, effectively mitigating cross-modality discrepancies.

### 4.4 Further Analysis

**Hyper-parameter Analysis.** There are three key hyper-parameters in our method, and we give the quantitative results to evaluate their influence with different values in Fig. 4. As we can see, the best performance is achieved when $\alpha$ is set to 0.5, $\beta_1$ is set to 0.5, and $\beta_2$ is set to 10.0, respectively.

**Accuracy of Pseudo-labels.** As shown in Fig. 5, we compared our method with several SOTA USVI-ReID methods on four common clustering evaluation metrics to show the effectiveness of the proposed RPNR. The results show that the visible and infrared pseudo-labels generated by RPNR significantly outperform existing methods on all four clustering metrics, indicating that our method provides more reliable pseudo-labels for network training, thereby boosting performance improvement.

**Visualization Analysis.** We visualize the visible and infrared feature distribution with t-SNE in the 2-D embedding space, which contains 10 randomly selected identities. As shown in Fig. 6, compared to the Baseline, in our approach, the feature distributions

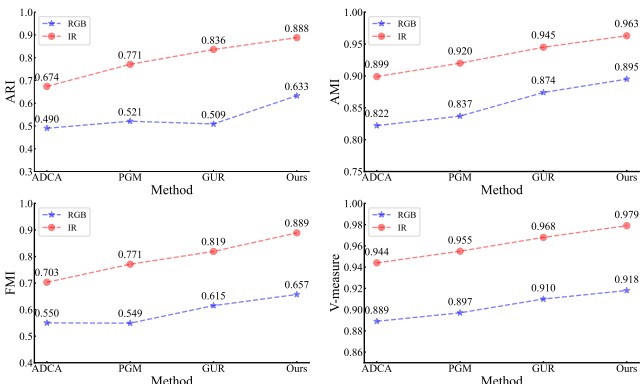

**Figure 5: Four clustering evaluation metrics compared with SOTA methods on the SYSU-MM01 dataset. "RGB" and "IR" denote the accuracy of visible and infrared pseudo-labels.**

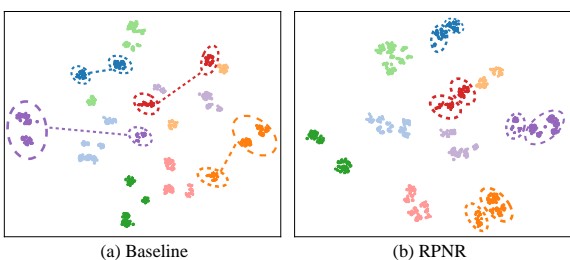

**Figure 6: The t-SNE visualization of randomly chosen 10 identities, with each identity represented by a distinct color and each modality denoted by different shapes.**

of the same identities from the same modality are more compact (see orange and purple circles), and the feature distributions of the same identities from different modalities are also closer (see red and blue circles). This indicates that RPNR significantly reduces cross-modality disparities and establishes a solid foundation for reliable cross-modality correspondences.

### 5 Conclusion

In this paper, we introduce an effective approach for addressing the USVI-ReID task, termed Robust Pseudo-label Learning with Neighbor Relation (RPNR). Our goal is to explore more reliable pseudo-labels and establish more dependable cross-modality correspondences for the USVI-ReID task. To this end, we first employ the Noisy Pseudo-label Calibration module to rectify noisy pseudo-labels, thereby obtaining more reliable pseudo-labels. Subsequently, we present the Neighbor Relation Learning module to model the potential interactions between different samples. In addition, we introduce the Optimal Transport Prototype Matching module to establish dependable cross-modality correspondences at the cluster level. Finally, we propose the Memory Hybrid Learning module to mine modality-specific and modality-invariant information while mitigating significant cross-modality disparities. Comprehensive experimental results on two popular benchmarks demonstrate the effectiveness of the proposed method.

## Acknowledgments

This work is supported by the National Natural Science Foundation of China (No. 62176224, 62222602, 62106075, 62176092, 62306165, 62376233), Natural Science Foundation of Shanghai (23ZR1420400), Natural Science Foundation of Chongqing (CSTB2023NSCQ-JQX0007), China Postdoctoral Science Foundation (No. 2023M731957), CCF-Lenovo Blue Ocean Research Fund, open project of China Academy of Railway Sciences (No. 2023YJ357), and in part by Xiaomi Young Talents Program.

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
