# OpenReview forum: "Robust Pseudo-label Learning with Neighbor Relation for Unsupervised Visible-Infrared Person Re-Identification"
_acmmm.org/ACMMM/2024/Conference — MM2024 Poster_

### Official Review · Reviewer_Mzpy · 2024-04-28

**Rating:** 5
**Confidence:** 2

**Summary:**

This paper centers on addressing the unsupervised VI-ReID problem. It introduces a Noisy Pseudo-label Calibration (NPC) method aimed at rectifying noisy intra-modality labels based on the relationships between instances and prototypes. Additionally, a Neighbor Relation Learning (NRL) module leverages pairwise instance-level relationships to enhance intra-modality learning. An Optimal Transport Prototype Matching (OTPM) module is proposed to establish cluster-level associations by formulating an optimal transport problem. Furthermore, a Memory Hybrid Learning (MHL) module is introduced to perform both modality-specific and modality-agnostic contrastive learning, facilitating the acquisition of modality-invariant features.

**Strengths:**

(1) The proposed method achieves remarkable performance on two public datasets, surpassing the state-of-the-art approaches by a considerable margin.\
(2) The sufficient experiments demonstrate the effectiveness of each component of the method.

**Limitations:**

(1) The OTPM module primarily addresses the inconsistency in the number of clusters across different modalities in PGM [1]. While [2,3] propose bi-directional strategies and [4, 5] directly transfer infrared labels to visible instances to tackle this issue, the superiority of OTPM over these methods is not clearly elucidated and demonstrated compared to the above methods.\
(2) The hybrid memory strategy is widely employed in existing methods [1,2,4,5], and the loss terms in Eq. 22 appear to stem from the MSMA framework [2]. The authors are suggested to analyze the differences between MHL and MSMA.\
(3) The NPC and NRL modules focus on addressing intra-modality noisy labels, offering limited insights into cross-modality learning.\
(4) The proposed method integrates existing technology (ACCL [1]); however, the performance enhancement attributed to ACCL is not delineated in the paper.

[1] Wu Z, Ye M. Unsupervised visible-infrared person re-identification via progressive graph matching and alternate learning[C]//Proceedings of the IEEE/CVF Conference on Computer Vision and Pattern Recognition. 2023: 9548-9558.\
[2] Cheng D, He L, Wang N, et al. Efficient bilateral cross-modality cluster matching for unsupervised visible-infrared person reid[C]//Proceedings of the 31st ACM International Conference on Multimedia. 2023: 1325-1333.\
[3] Cheng D, Huang X, Wang N, et al. Unsupervised visible-infrared person reid by collaborative learning with neighbor-guided label refinement[C]//Proceedings of the 31st ACM International Conference on Multimedia. 2023: 7085-7093.\
[4] Yang B, Chen J, Ye M. Towards grand unified representation learning for unsupervised visible-infrared person re-identification[C]//Proceedings of the IEEE/CVF International Conference on Computer Vision. 2023: 11069-11079.\
[5] Yang B, Chen J, Chen C, et al. Dual Consistency-Constrained Learning for Unsupervised Visible-Infrared Person Re-Identification[J]. IEEE Transactions on Information Forensics and Security, 2023.

**Suitability:**

3

---

### Official Review · Reviewer_d627 · 2024-04-29

**Rating:** 4
**Confidence:** 4

**Summary:**

This paper proposes an unsupervised RGB-IR person ReID. It introduces neighbor relation-based pseudo-label calibration to get robust pseudo-labels and reduce intra-class variations, then applies optimal transport matching to build cross-modal correspondence. Moreover, it proposes a memory hybrid module to learn modality-specific/invariant information. Experiment results indicates the superior of the proposed method.

**Strengths:**

1. The paper overall is easy to follow, and the proposed method is presented in detail.
2. This paper considers both the noisy pseudo-labels and cross-modal association problems presented in the USVI-ReID task, which is interesting and realistic.
3. Comprehensive experimental results show that the proposed method surpasses the state-of-the-art methods and gains promising performances.

**Limitations:**

1. The experiment on LLCM [R1] should be included.
2. The whole method looks similar to PGM, OTLA, and SPCL.  What is the key difference between the proposed method and the aforementioned methods?

[R1] Diverse Embedding Expansion Network and Low-Light Cross-Modality Benchmark for Visible-Infrared Person Re-identification， CVPR 2023

**Suitability:**

3

---

### Official Review · Reviewer_rCCZ · 2024-05-26

**Rating:** 4
**Confidence:** 3

**Summary:**

This work is aimed at learning robust features for unsupervised visible-infrared person re-identification (USVI-ReID). To this end, the authors design robust pseudo-label generation, intr- and inter-modality feature learning methods, improving the overall learning procedure of USVI-ReID.

**Strengths:**

The proposed method mainly improves the training procedure without designing extra trainable modules and adding no extra cost to the inference. Both the generation of pseudo-labels and the learning of robust multi-modal person features are carefully designed. The overall USVI-ReID performance is shown to be superior. Analytical experiments are conducted to help understand the effectiveness of the proposed method.

**Limitations:**

1. While the issue to be solved in this work is claimed to be "calibrate noisy pseudo-labels usually associated with hard samples" (as in lines 15-17), the proposed Noisy Pseudo-label Calibration method in fact re-generates the integral pseudo-labels rather than specifically refining those of the hard samples. Neither specific experiments nor in-depth discussions are presented to demonstrate the effectiveness of the proposed NPC on those hard samples.

2. Why is the Jaccard Similarity necessary in NPC? Is it practical to employ other similarity metrics, e.g. cosine similarity?

3. Basically, Neighbor Relation Learning is to employ a previously proposed metric loss for intra-modality feature learning. How about other simpler losses, e.g. triplet loss? Any discussion about the innovation/difference of NRL compared with previous works?

4. OTPM is said to be "following PGM and OTLA", while this module is claimed to be a contribution, the differences between OTPM and PGM/OTLA are not discussed. Although Figure 3 shows the superior accuracy of cross-modality correspondence compared with PGM, more technical details should be given to see if the improvement comes from the better-aligned features rather than the cross-modal matching algorithm.

**Suitability:**

3

---

### Meta-Review · Area_Chair_bhVh · 2024-07-01

**Recommendation:** Accept (Poster)
**Confidence:** 4

**Metareview:**

All reviewers agree on the superior and comprehensive performance of the proposed in this paper. Other merits include satisfying writing and idea novelty. In addition, the rebuttal has well addressed concerns from reviewers. After rebuttal, all reviewers have made positive recommendation of this paper towards acceptance.